# Plant-Derived Smoke Solution Alleviates Cellular Oxidative Stress Caused by Arsenic and Mercury by Modulating the Cellular Antioxidative Defense System in Wheat

**DOI:** 10.3390/plants11101379

**Published:** 2022-05-22

**Authors:** Muhammad Ibrahim, Sadam Nawaz, Khalid Iqbal, Shafiq Rehman, Riaz Ullah, Ghazala Nawaz, Rafa Almeer, Amany A. Sayed, Ilaria Peluso

**Affiliations:** 1Department of Botany, Kohat University of Science and Technology, Kohat 26000, Pakistan; mi680310@gmail.com (M.I.); nawazsadamwazir@gmail.com (S.N.); khalidiqbalkhan5560@gmail.com (K.I.); 2Department of Biology, University of Haripur, Haripur 22620, Pakistan; drshafiq@uoh.edu.pk; 3Department of Pharmacognosy, College of Pharmacy, King Saud University, P.O. Box 2455, Riyadh 11451, Saudi Arabia; rullah@ksu.edu.sa; 4Department of Zoology, College of Science, King Saud University, P.O. Box 2455, Riyadh 11451, Saudi Arabia; ralmeer@ksu.edu.sa; 5Zoology Department, Faculty of Science, Cairo University, Giza 12613, Egypt; amanyasayed@sci.cu.edu.eg; 6Research Centre for Food and Nutrition, Council for Agricultural Research and Economics (CREA-AN), 00178 Rome, Italy; i.peluso@tiscali.it

**Keywords:** ROS, antioxidative system, PDS, arsenic and mercury stress, stress alleviation

## Abstract

Heavy metal stress is a significant factor in diminishing crop yield. Plant-derived smoke (PDS) has been used as a growth promoter and abiotic stress alleviator for the last two decades. Although the roles of PDS have been determined in various plants, its role in ameliorating heavy metal stress in wheat has not been reported so far. Therefore, the present work was conducted to investigate the effect of smoke solution extracted from a wild lemongrass *Cymbopogon jwarancusa* (*C. jwarncusa*) on physiological and biochemical features of wheat under arsenic (As) and mercury (Hg) stress. The results showed that higher concentrations of As and Hg pose inhibitory effects on wheat seed germination and seedling growth, including shoot/root length and shoot/root fresh weight. Photosynthetic pigments, such as chlorophyll a and b and carotenoids, were significantly decreased under As and Hg stress. Importantly, the levels of H_2_O_2_, lipid peroxidation, and TBARS were increased in wheat seedlings. The activity of antioxidant enzymes, such as CAT, was decreased by As and Hg stress, while the levels of SOD, POD, and APX antioxidant enzymes were increased in root and shoot. Interestingly, the application of PDS (2000 ppm), individually or in combination with either As or Hg stress, enhanced wheat seed germination rate, shoot/root length, and shoot/root fresh weight. However, the levels of H_2_O_2_, lipid peroxidation, and TBARS were decreased. Similarly, the levels of SOD, POD, and APX were decreased by PDS under As and Hg stress, while the level of CAT was enhanced by PDS under As and Hg stress. Interestingly, the levels of chlorophyll a and b, and total carotenoids were increased with the application of PDS under As and Hg stress. It is concluded that PDS has the capability to alleviate the phytotoxic effects of As and Hg stress in wheat by modulating the antioxidative defense system and could be an economical solution to reduce the heavy metal stress in crops.

## 1. Introduction

Heavy metal contamination is one of the environmental threats diminishing plant productivity. Most of the anthropogenic activities and rapid industrialization are the sources of heavy metals in the natural environment [1]. Heavy metals are required in traces for plants and animals to survive; however, their high concentrations may inhibit growth and metabolic processes. Among the heavy metals, some metals, such as Hg, etc., are harmful to plant growth, even in small traces. Moreover, these harmful heavy metals are transported and accumulated in crops and, thus, cause damage to plants and animal health [2]. As is one of the growing concerns in metal and metalloids, due to its highly toxic nature and naturally abundant nature in our climate [3]. As has been reported in rocks, soils, natural bodies of water, and also in living organisms. In the environment, 60% of As has resulted due to large anthropogenic activity [4]. As pollutes soil and groundwater, allowing it to leach out of the soil and enter the food chain, resulting in serious threats to plants and animals [5]. The ability of plants to transport As and other hazardous heavy metals and metalloids are mostly observed by the bioavailability of As rather than the amount of As in the soil [6,7]. The use of such soil and water polluted by As for the purpose of irrigation greatly affects the yield, fresh weight, and shoot/root length of the plants [8]. The elevated As concentration in cultivated land can cause phytotoxicity in crop plants, resulting in shrunken stomata, yellow leaves, suppressed roots, and reduced plant growth and yield [9]. The intake of non-essential elements, including As, by plants can interfere with the uptake of other critical nutrients in the soil, such as phosphorus, through phosphate transport systems [10].

Hg is a significant contaminant in soils [11]. It is emitted after coal-fired power stations’ storage tanks have been depleted, which causes Hg pollution in the surrounding areas. Hg can be found in both gaseous and particle forms in the air [12,13]. The suspended particles of Hg in the air fall on the surface of leaves and can be absorbed by roots [14,15]. Only a small amount of Hg is discharged into the environment or moved to other parts of the plant body, and the rest of it persists in the plant body, with little movement [14,16]. Hg is one of the most toxic and poisonous metals, even in extremely low quantities, due to its tendency to accumulate in living beings [17]. Certain metabolic activities in plants, such as water absorption and nitrogenous chemical production, are inhibited by the uptake of Hg [18]. Increased concentration of Hg in agricultural land can decrease the productivity of plants. However, low concentrations of Hg can greatly affect plant essential activities, such as seed germination, absorption of water, enzymatic activities, photosynthetic pigments, cell division, and mineral nutrition [19]. Hg has been shown to affect the gating of aquaporins in the epidermal cell of *Allium cepa*, due to which, the absorption of water uptake was disturbed [20]. Hg has been shown to have an adverse influence on the antioxidant protection mechanism by interacting with non-protein thiols (NPSH) and non-enzymatic antioxidants, such as glutathione (GSH), as well as enzymatic antioxidants, such as ascorbate peroxidase (APX), superoxide dismutase (SOD), and glutathione reductase (GR) [21,22,23].

Seed germination and development of plants is an effective process that can be affected by various abiotic factors, such as water, light intensity, temperature, oxygen, and heavy metals [24].

Plant-derived smoke (PDS) is a complex chemical mixture of plant-active chemicals that can stimulate a wide range of plant species in a given ecosystem. PDS has been reported to play an important role in the seed germination of various endangered plant seeds [25]. PDS has also been demonstrated to promote seed propagation, seedling development, and crop vigor, in a wide variety of horticulture and agricultural crops [26]. Reportedly, PDS has enhanced the seed germination rate by increasing seed exposure to endogenous GA_4_ [27]. It has been demonstrated that PDS plays a key role in the alleviation of abiotic stresses. Seed primed with PDS has been found to increase seed germination percentage and seedling fresh weight in salt-stressed maize [28,29,30]. *Sapium sabiforum* has shown improved tolerance to salt and osmotic stresses by increasing germination and post-germination growth in *PDS* [31]. Under heat and osmotic stress [32], karrikins, the active compound in PDS, improved seed germination and seedling growth in *Eragrostis tef* (zucc.) via regulating GA biosynthesis-related genes, enhancing ROS-scavenging antioxidant enzymes and sugar uptake in seeds [33,34,35]. Abiotic stress conditions have been shown to have a deleterious impact on wheat morphological, physiological, and biochemical properties [36].

Wheat is one of the most frequently farmed cereal crops in the world, with 40 percent of the planet’s population dependent on it for nourishment [37]. The global population is predicted to grow to 8 billion people by 2030, making food supply a serious issue [38]. Abiotic stress challenges have been shown to have a deleterious impact on wheat morphology, physiology, and biochemical properties [39].

As and Hg stress affect several physiological processes in wheat plants, particularly crop growth and seed germination. Heavy metals cause an increase in reactive oxygen (ROS) molecules, such as H_2_O_2_, O^−2^, and OH, resulting in severe cell damage [40,41].

The promontory effect of PDS occurs at various growth and developmental levels in plants, and the present study investigated how PDS affects the growth of wheat under As and Hg stress. Certain morphological, physiological, and biochemical activities of wheat were measured. The data showed that PDS can enhance wheat growth under As and Hg stress conditions by scavenging ROS, and modulating antioxidant enzyme activity. PDS was used at 2000 ppm concentration, which enhanced the germination and post-germination growth of wheat seedlings. It alleviated the negative effects of As and Hg on germination and post-germination growth. Our research demonstrates that PDS could be an economical plant product to enhance wheat physiological and biochemical processes under As and Hg stress, which are essential for the induction of tolerance in wheat.

## 2. Materials and Methods 

### 2.1. Preparation of Solutions

*Cymbopogon jwarancusa* L. (*C. jwarancusa*), also locally known as Sargara, was collected from the premises of Kohat University of Science and Technology. *C. jwarancusa* plants were thoroughly washed with water and shade dried for 15 days. PDS solution was prepared by weighing 333 gm of the semi-shade-dried leaves of *C. jwarancusa* which were burned in a locally constructed furnace on an electric heater [42,43]. The furnace’s window was sealed with aluminum foil to reduce the possibility of smoke leakage. An emission pipe with a furnace for directing smoke toward water ended in a beaker filled with distilled water. This made it easier for the smoke to bubble in the distilled water. The resulting solution was called a stock solution which was further diluted to 2000 ppm for further use in the experiment. Previously, we screened several PDS dilutions such as 500, 1000 and 2000 ppm for the alleviation of salt stress [43]. PDS 2000 ppm significantly alleviated the salt stress and during present experiments, PDS 2000 ppm performed better. Therefore, in the present study, the experiments were carried out at 2000 ppm. The PDS stock solution was stored at −20 °C for subsequent use. As and Hg solutions were prepared using As_2_O_3_ and HgCl_2_.

### 2.2. Seed Source and Sterilization

The seed variety Paseena, 2017 was used in the experiments. Healthy and uniform-sized seeds were selected and surface sterilized with 0.1% sodium hypochlorite. Next, the seeds were washed with 70% ethanol for 3 min and rinsed with distilled water five times each for 1 min. Seed germination test was conducted thrice which revealed 100% germination, indicating that the wheat seeds were good in health and had the ability to germinate. Seeds of similar size were then chosen and germinated on Petri plates (150 × 60 × 100 mm) containing double-layered Whatman No. 1 filter paper. Before the experiment, the Petri plates and filter paper were pre-sterilized. Each petri dish had twelve seeds placed on it, and three Petri dishes were used for each treatment. Seeds were treated with PDS 2000 ppm, As and Hg 2 mM solution and distilled water, which were considered as controls. All Petri plates were kept at 25 degrees Celsius for 16 h of light and 7 h of darkness at 60% humidity. After 24 h, the rate of germination was measured. Germinated seeds are those that have a protruding radical. When the clear differences were shown on day 7 of germination, the morphological parameters such as shoot/root length and fresh weight of the seedling were measured. All experiments were performed independently three times.

### 2.3. Measurement of H_2_O_2_ and Lipid Peroxidation

To measure the levels of H_2_O_2_ in the cell, under treatment of PDS, As, Hg, As + PDS and Hg + PDS, a reaction mixture of 1 mL of enzyme extract, 1 mL of 10 mM potassium phosphate buffer (PH 7.0), and 2 mL of 1 M sodium phosphate buffer (pH 7.0) was prepared and the absorbance was measured at 390 nm following the method described previously [44]. Spectrophotometric analysis of potassium iodide (KI) was performed. The level of lipid peroxidation was assessed using thiobarbituric acid reactive substances (TBARS) in the presence of PDS, As, Hg, PDS + As, and PDS + Hg. On day 7, 0.25 g of the seedlings were collected and pulverised in a mortar with the use of a pestle, using 3 mL of 25% thiobarbituric acid (TBA) and 10% trichloroacetic acid (TCA) [45]. After being heated for 30 min at 95 °C, the homogenate was centrifuged for 10 min at 10,000× *g* before being immediately chilled in an ice bath. The absorbance was measured at 532 nm, and non-specific turbidity was determined at 600 nm.

### 2.4. Quantification of Antioxidant Enzymes

Seedlings were harvested at day 7 following germination to assess antioxidant enzyme activity. As such, 500 mg of fresh wheat seedlings was grounded in a pre-cold mortar and pestle and homogenized in a potassium phosphate (K_2_HPO_4_._2_H_2_O) buffer (pH 7.8) (10 mL out of 50 mM). The samples were centrifuged at 4 °C for 20 min at 12,000 rpm after chilling. The supernatant was collected after centrifugation for additional enzymatic examination. After being heated for 30 min at 95 °C and immediately cooled in an ice bath, the homogenate was centrifuged for 10 min at 10,000× *g* for 10 min.

#### 2.4.1. Measurement of Superoxide Dismutase (SOD) Activity

To determine the SOD, 250 mL of the total reactive homogeneous mixture was used (Nitroblue tetrazoilum chloride 15.5 mg, methionine 485 mg, EDTA 10 mg, and Riboflavin 0.02 mg). After being prepared and wrapped in aluminium foil, the reaction substrate was maintained in the refrigerator to prevent degradation. Next, 2.725 mL of the homogeneous mixture, 25 mL enzyme extract, and 25 mL H_2_O_2_ were used to assess SOD, as well as a blank sample (without enzyme extract) exposed to light (4000 lux for 20 min).

Instead of enzyme extract, pure water (25 mL) was used as a control sample. When the combination was exposed to lux, the tubes were wrapped in black cloth for a steady response. Glass cuvettes were used to quantify the samples at 560 nm [46].

#### 2.4.2. Quantification of Peroxidase (POD) and Ascorbate Peroxidase (APX) Activity

Using the procedures described previously [47], the activities of POD and APX were assessed. The reaction sample contained enzyme extract, 1.5 percent guaiacol, and H_2_O_2_ in quantities of 100 mL each. PBS (2.8 mL), guaiacol (0.1 mL), and H_2_O_2_ were included in the control sample. The POD activity was determined using a spectrophotometer with an absorbance wavelength of 470 nm. Potassium phosphate, EDTA-Na_2_, H_2_O_2_, ascorbic acid, and enzyme extract were used to make the buffer. The utilized wavelength was 290 nm.

#### 2.4.3. Statistical Evaluation

The findings from each test were examined after three repetitions. All of the data were displayed as an average standard deviation using the statistics 9 applications (Analytical Software, 2105 Miller Landing Rd, Tallahassee, FL 32312, and version 8.1, Tallahassee, FL, USA). One-way analysis of variance (ANOVA) was used to establish the significance of variations between the treatment and control groups, with (*p ≥* 0.05) being the least significant variation. The samples were compared using the LSD test.

## 3. Result and Discussions

### 3.1. PDS Promotes Wheat Seed Germination under As and Hg Stress 

Seed germination plays a key role in the life cycle of a plant. To determine how PDS affects seed germination under As and Hg stress, wheat seeds were treated with As and Hg alone and combined with PDS. The results indicated that the individual treatment of As and Hg stress caused a reduction in germination. The results showed that seeds treated with As showed 23.33% germination after 24 h, 33.33% after 48 h, and 40% after 72 h. Similarly, treatment of Hg resulted in 28% germination at 24 h, 40% at 48 h, and 43.33% at 72 h. This germination result was clearly comparable to that of seeds treated with distilled water, which showed 52% germination at 24 h, 63.33% at 48 h, and 93.33% at 72 h. On the other hand, seeds treated with PDS (2000 ppm) showed 63.33% germination at 24 h, 75.55% at 48 h, and 100% at 72 h. These results indicate that both As and Hg have inhibitory effects while PDS has a promoting effect on seed germination. Next, we wanted to know whether PDS can alleviate the inhibitory effects of As and Hg. We treated wheat seeds with As or Hg solutions supplemented with PDS. The result showed an increase in seed germination by 53.33%, 73.33%, and 86.66% under As stress, and 56.66%, 76.66%, and 86.66% under Hg stress at 24, 48, and 72 h, respectively (Figure 1). The results indicate that PDS (2000 ppm) promotes seed germination by alleviating the adverse effects of As and Hg at the germination level. The results also demonstrate that compared to Hg, the As effect was mild at germination level, as indicated by Figure 1.

Soil pollution from heavy metals is a major environmental problem that limits crop productivity. As and Hg are toxic chemicals that injure plants. As and Hg have adverse effects on plant growth and development. Likewise, the majority of biochemical and physiological processes also get disturbed because of toxicity [48,49,50,51]. Recently, various concentrations of PDS have been used to restore plant growth under salt stresses; however, of these concentrations, PDS (2000 ppm) was found to most significantly alleviate the negative effect of salt stress [43]. Looking at the previous findings with PDS 2000 ppm, in the present study, we used PDS 2000 ppm to counteract the inhibitory effects of As and Hg stress, which is considered an ultimatum to the environment. When seedlings were exposed to As and Hg stress, germination was significantly reduced, as compared to control. Previous findings suggest that metallic stress creates osmotic pressure when seedlings are unable to ingest water, resulting in reduced seed rate germination during As and Hg stress [52]. Heavy-metal-induced osmotic stress can be reduced and seed water absorption can be increased with smoke water [53]. In the present findings, PDS increased wheat seed germination. Previously, Karrikinolide, an active ingredient in the smoke solution, has been shown to play a role in accelerating the development of radicals in seed germination by stimulating cell cycle events [54]. Our findings are also backed up by Malook [55], who concluded that PDS has the capacity to enhance the seed germination rate. Altogether, these results conclude that PDS has the capability of alleviating heavy metal stress in crop species.

### 3.2. PDS Enhances Wheat Seedling Growth under As or Hg Stress 

Next, the effect of As and Hg was evaluated on wheat seedlings, and the treatments resulted in reduced wheat shoot length by 4.8 and 4.26 cm under As and Hg stress, as compared to control, which was 9.2 cm. On the contrary, individual PDS treatment enhanced shoot length by 14.8 cm. Interestingly, As and Hg treatment followed by PDS treatment enhanced the shoot length to 7. 85 and 7.2 cm, respectively (Figure 2A). In case of root, the root length was reduced to 5 and 4.8 cm by treating As and Hg, respectively, while PDS increased the root length by 15 cm compared to control, which was 13 cm (Figure 2B). Quantitative analysis of root with As + PDS showed 8.1 cm length, while root length with Hg + PDS was measured as 7.8 cm (Figure 2B). Next, measurements of wheat root and shoot fresh weights were taken. The data showed that As and Hg stress reduced shoot fresh weight to 0.5 and 0.59 gm, respectively, while PDS increased shoot fresh weight to 0.81 gm compared to control (0.69 gm). Supplementation of As and Hg with PDS removed the inhibitory effects and increased shoot fresh weights to 0.7 and 0.71 gm (Figure 2C). Root fresh weight was analyzed under individual treatment of As and Hg each and resulted in 0.55 and 0.52 gm, as compared to control (0.9 gm). PDS treatment enhanced root fresh weight by 1 gm and, in combination with As and Hg, also enhanced the root fresh weight by 0.79 and 0.71 gm, respectively (Figure 2D). Photographs shown in Figure 2E represent the effect of As, Hg, and PDS alone and in combined form. These results indicate that As and Hg reduced wheat seedling growth, while PDS increased the seedling growth in individual treatment, as well as in combined form with As and Hg.

The principal secondary organs of a plant are its shoots and roots. Increased seedling biomass shows that the seedlings are growing and developing rapidly [23]. In the present study, the subjection of wheat seedlings to As and Hg stress resulted in shortened leaves and roots, as well as reduced seedling fresh and dry weight. Our results are supported by previous findings, which state that wheat seedlings subjected to As and Hg stress reduce the lengths of branches and roots [56]. Hg has been reported to reduce coleoptile and root growth in various species of brassica [57]. Altogether, these results prove that heavy metals, particularly As and Hg, negatively affect plant growth and biomass. To alleviate the As and Hg-induced inhibitory effects, PDS was used in combination with As and Hg. The results indicated that PDS restored seed germination, seedling growth and root/shoot fresh mass. A detailed study regarding the mitigatory role of PDS on the alleviation of negative effects of abiotic and heavy metal stress indicated that PDS plays a significant role in enhancing plant growth under heavy metal stress (Shabbir and Ilyas 2019, book chapter). Recently, PDS has been reported to mitigate the toxic effects of salt stress in wheat germination, seedling growth, and seedling fresh biomass [43]. Another study determined the well-proven alleviation effects of PDS on salt stress and improved the seed germination rate in maize [27]. Karrikins, the PDS-derived active compounds, have resulted in improved seed germination and root and shoot length in *Eragrostis tef* (zucc.) under osmotic stresses and heat stress [31]. Karrikins have been shown to increase seed germination in Arabidopsis, maize, and beans via modulating the expression of GA biosynthesis-related genes, boosting ROS-scavenging antioxidants, and sugar mobilization [58]. Collectively, these studies suggest that PDS could play an essential role in ameliorating the toxic effects of heavy metal stress in crop species. 

As and Hg stress primarily affect plant physiological parameters, such as shoot and root vigor, while PDS alleviates As and Hg stress. Smoke water dramatically increased seedling growth, including shoot and root lengths, in all plants. This could be due to butenolide’s ability to stimulate root growth [59,60]. Light et al. [61] and Brown et al. [62] both found that smoke improves plant vigor and that the advantages of smoke extend beyond germination improvement to increase seedling vigor. When young sprouts were exposed to smoke, their vegetative growth was healthier [63]. According to Sparg et al. [64], smoke-treated seeds showed higher potency, and seedlings were taller and heavier than untreated seeds. In the current study, wheat seeds subjected to As and Hg stress showed significantly lower seedling biomass than controls. However, when wheat seeds were subjected to As and Hg, in addition to PDS, the seedling biomass was enhanced compared to As and Hg stress alone, but was reduced compared to control. PDS, as previously indicated, helps the plant body produce more leaves, resulting in an increase in plant biomass [65]. It is evident from the previous and present study that PDS could mitigate the inhibitory effect of heavy metal stress on crop biomass and it is likely that PDS may have promoting effects on crops yield under heavy metal stress conditions.

### 3.3. Combined Effect of As, Hg Stress and PDS (2000 ppm) Solution on Photosynthetic Pigments of Wheat

To understand whether As and Hg PDS affect photosynthetic pigments in wheat seedlings, 12-day-old leaves of wheat seedlings from each treatment were analyzed for chlorophyll a and b contents. The results indicated that chlorophyll a and b content in wheat seedlings, exposed to PDS (2000 ppm) and As and Hg alone, showed significant variations. The obtained results showed that seedlings treated with PDS (2000 ppm) possessed high chlorophyll a (7.284 µg/mg FW) and b (5.547 µg/mg FW) compared to a (6.43 µg/mg FW) and b (4.506 µg/mg FW) in control seedlings. On the contrary, exposure to As showed a significant decrease in chlorophyll a (3.873 µg/mg FW) and b (3.196 µg/mg FW), while Hg stress caused a reduction in a and b to 2.659 and 2.442 µg/mg FW, respectively. To alleviate the phytotoxic effect of As and Hg through the use of PDS (2000 ppm), the levels of chlorophyll a and b were recorded. Notably, treatment of As + PDS and Hg + PDS had high chlorophyll a and b of 5.4567 and 3.974 µg/mg FW and 5.387 and 3.984 µg/mg FW, respectively, compared to seedlings treated with only As and Hg (Figure 3). These results show that PDS can reduce the inhibitory effect of As and Hg on chlorophyll a and b content in wheat leaves. Analysis of total carotenoids showed a significant decrease in the plant exposed to As and Hg stress. Seeds treated with PDS (2000 ppm) showed a significant increase in carotenoids compared to control. The results indicate that total carotenoids in plants treated with control, PDS (2000 ppm), As, Hg, As + PDS, and Hg + PDS were 1.986, 2.242, 0.829, 1.298, 1.869, and 1.753 µg/mg FW, respectively. The results indicate that the phytotoxic effect of As and Hg was significantly alleviated by PDS (2000 ppm) on photosynthetic pigments (Figure 3).

Photosynthetic pigments are the most fundamental and basic ingredients in the photosynthesis process and, hence, play a crucial part in the creation of plant food. It is created by plants throughout the production cycle to meet their nutritional requirements. In the current study, As and Hg toxicity resulted in a considerable reduction in chlorophyll and carotenoids production in wheat leaves. These findings support prior studies that revealed a reduction in chlorophyll concentration in *Brassica napus* as a result of Hg and As exposure [66]. Heavy-metal-induced reduction in photosynthetic pigments can be caused by the inhibition of enzymes responsible for chlorophyll production, such as protochlorophyllide reductase and -aminolaevulinic acid dehydratase (ALA-dehydratase). Interestingly, the supplication of PDS with As and Hg stress significantly enhanced chlorophyll a and b and carotenoid contents in wheat leaves. In several plant species, such as maize, PDS has been proven to boost photosynthetic pigments, such as chlorophyll a and b, and carotenoids [67,68]. 

### 3.4. Effect of PDS on H_2_O_2_ and TBRS under As and Hg Stress

The levels of H_2_O_2_ in the roots and shoots of wheat seedlings were measured. Treatment of wheat seedlings with As and Hg stress caused high levels of H_2_O_2_ in both root and shoot, as compared to control plants. To find the level of oxidative stress in different seedling parts, levels of H_2_O_2_ were measured in shoots and roots individually. The results indicated that the content of H_2_O_2_ in the root was 0.709 and 0.691 µM/g FW under As and Hg-treated seedlings, respectively, while the shoot was found to have 0.614 and 0.590 µM/g FW levels of H_2_O_2_ in As and Hg-treated seedlings, respectively (Figure 4A,B). These values were comparable to the control seedlings, which were 0.4 µM/g FW in shoots and roots, respectively. PDS-treated shoots and roots had 0.4 and 0.39 µM/g FW levels of H_2_O_2_, which were lower than control seedlings. The results also indicated that PDS can decrease the H_2_O_2_ content produced in roots and shoots (0.344 and 0.397 µM/g FW, respectively), while in control, it was 0.403 and 0.403 µM/g FW, respectively. When the PDS was used in combination with As and Hg, the content of H_2_O_2_ in roots reduced to 0.462 and 0.453 µM/g FW, while in the shoots, it decreased to 0.515 and 0.504 µM/g FW, respectively. The results indicate that PDS 2000 ppm has the capability to reduce H_2_O_2_ concentration under As and Hg stress.

To determine how As, Hg, and PDS affect TBARS levels in wheat, we treated wheat seeds with As, Hg, or PDS alone. The results showed that individual treatment of As and Hg enhanced the levels of TBARS in the roots and shoots, as compared to control. Quantitative data for As-treated roots and shoots showed 0.515 and 0.526 µM/g FW levels, while 0.497 and 0.521 µM/g FW for Hg-treated roots and shoots, respectively. The levels of TBARS in both roots and shoots under As and Hg for control were 0.376 and 0.269 µM/g FW, respectively. Plants treated with PDS 2000 ppm had low levels (0.329 and 0.247 µM/g FW) compared to control. When seedlings were treated with PDS 2000 ppm in a combined solution with As and Hg, the levels of TBARS were significantly reduced in roots/shoots to 0.413 and 0.368 µM/g FW for As and 0.401 and 0.356 µM/g FW for Hg, respectively (Figure 4C,D). The results indicate that TBARS concentration in the shoots was higher than in the roots, which means that the shoot is more affected by As and Hg stress.

Heavy metals, such as metalloids, have been shown to contribute to oxidative stress by promoting the generation of free radicals and reactive oxygen species [69]. In the current study, As and Hg-stressed wheat plants produced more H_2_O_2_ and TBARS. Seedlings exposed to As and Hg stress had significantly higher levels of H_2_O_2_ and TBARS, while those treated with PDS showed a drop in H_2_O_2_ and TBARS contents. These results are in line with those that found that As, Hg, and Ni toxicity increased H_2_O_2_ and MDA contents in *Glycine max*, maize, and Allium species [70,71,72]. A recent study showed high levels of H_2_O_2_ and TBARS under salt stress, while the application of PDS reduced the levels of H_2_O_2_ and TBARS under salt stress [43]. These findings conclude that PDS has the capability of ameliorating As and Hg stress by reducing the oxidative stress in plant cells. 

### 3.5. PDS Reduces Levels of Lipid Peroxidation Production under As and Hg Stress

The level of lipid peroxidation was determined and the results indicated the highest level of lipid peroxidation when wheat seedlings were treated with As and Hg. Quantitative analysis showed that the level of lipid peroxidation in roots was 25.28 µM/g FW and 24.81 µM/g FW under As and Hg stress, while in shoots, it was 25.65 µM/g FW and 25.09 µM/g FW, respectively. When the plant was treated with PDS, the level of lipid peroxidation in the root and shoot was 12.96 µM/g FW and 13.68 µM/g FW, and in control, it was 18.37 µM/g FW and 18.47 µM/g FW, respectively. These results showed that PDS has the ability to decrease the level of lipid peroxidation in wheat seedlings. To alleviate the deleterious effect of As and Hg at lipid peroxidation level, PDS was used. The results showed that the lipid peroxidation in the root of wheat treated with As and Hg was 21.41 µM/g FW and 20.99 µM/g FW, while in the shoot it was 22.08 µM/g FW and 21.34 µM/g FW, respectively (Figure 4E,F). These results suggest that wheat treated with PDS inhibits lipid peroxidation production and protects the cell from the corrosive effect of lipid peroxidation.

### 3.6. PDS Modulates the Activity of Antioxidant Enzymes under As and Hg Stress

#### 3.6.1. PDS Lowers the Activity of Peroxidase (POD) under As and Hg Stress

The activity of POD enzymes was increased both in the roots and shoots of the wheat seedlings by 0.520 µM/g FW and 0.693 µM/g FW, respectively, when treated with As, while in Hg stress, it was found to be 0.514 µM/g FW and 0.635 µM/g FW for roots and shoots, respectively. The level of POD was recorded as 0.39 and 0.2.1 µM/g FW indicating lower levels of POD under PDS treatment compared to control. Combined application of PDS with As decreased POD level in roots and shoots by 0.319 µM/g FW, 0.470 µM/g FW, respectively, while in Hg + PDS, the activity of POD enzymes decreased to 0.315 and 0.479 µM/g FW in roots and shoots, respectively, as compared to control, which was 0.237 µM/g FW and 0.473 µM/g FW, respectively (Figure 5A,B).

POD is involved in a variety of mechanisms, such as lignin polymerization in the presence of H_2_O_2_, auxin catabolism, pectin cross linking, and growth regulation [73]. Under As and Hg stress, POD activity was elevated in wheat plants. Evidence has shown enhanced POD activity in wheat seedlings under heavy metal stress [74]. When maize (*Zea mays*) was exposed to arsenite and arsenate, similar outcomes were observed [75], which together with our results, suggest that the POD level increases in response to heavy metals and PDS reduces their levels. The lowering of POD level is likely due to the fact that phytotoxicity has been reduced. 

#### 3.6.2. PDS Decreases the Activity of Ascorbate Peroxidase (APX) under As and Hg Stress

In our findings, As and Hg increased the activity of APX in wheat seedlings. Briefly, the highest APX activity was found in the roots/shoots of seedlings treated with As and Hg, as compared to control. Quantification of roots and shoots treated with As resulted in 0.749 µM/g FW and 0.805 µM/g FW, while 0.732 µM/g FW and 0.805 µM/g FW with Hg. The lowest activity was recorded in roots (0.349 µM/g FW) and shoots (0.367 µM/g FW) treated with PDS compared to control (0.371 and 0.450 µM/g FW in root and shoot, respectively). Wheat seedlings treated with PDS in combination with As and Hg stress showed a decrease in APX activity compared to the seedlings treated with As and Hg treatment alone. In roots/shoots, the APX activity in As + PDS was 0.522 µM/g FW, and 0.573 µM/g FW, while in Hg + PDS it was 0.492 µM/g FW and 0.539 µM/g FW, respectively (Figure 5C,D). The above results indicate that PDS has the ability to alleviate the adverse effects of seedlings treated with As and Hg. Plants subjected to high concentrations of heavy metals, such as Ni, As, and Hg, have been reported to produce more APX in prior research [76]. Plants need APX to grow and develop properly [77]. H_2_O_2_ is a systemic intracellular signal that activates APX in response to abiotic stress [78]. Shri et al. [79] observed higher APX levels in rice seedlings under arsenic stress, which supports our findings.

#### 3.6.3. PDS Decreases the Activity of Superoxide Dismutase (SOD) under As and Hg Stress

The SOD enzyme was found to be increased as the concentrations of As and Hg increased. As compared to control, the activity of SOD was increased in the roots and shoots of the wheat seedlings under individual As and Hg stress. The wheat seedlings treated with PDS 2000 ppm had the lowest SOD enzyme in roots and shoots, recorded as 0.147 µM/g FW and 0.147 µM/g FW, while in control, it was 0.196 µM/g FW and 0.273 µM/g FW, respectively. When the plants were treated with As and Hg, the level of SOD enzyme in the roots and shoots was higher compared to control. The level of SOD enzyme in roots and shoots treated with As was 0.406 µM/g FW and 0.679 µM/g FW, while in Hg, it was 0.392 µM/g FW and 0.645 µM/g FW, respectively. The results indicate that PDS 2000 ppm has the ability to decrease the SOD enzyme. To alleviate the adverse effects of As and Hg on roots and shoots, the seedlings were treated with PDS 2000 ppm. It was observed that the level of SOD enzyme in root and shoot seedlings treated with As, in combination with PDS 2000 ppm, was decreased to 0.271 µM/g FW and 0.376 µM/g FW, while in Hg it was 0.264 µM/g FW and 0.363 µM/g FW, respectively (Figure 5E,F). The results indicate that the SOD enzyme activity was significantly alleviated by the use of PDS 2000 ppm.

The first enzyme of defense against oxidative stress is SOD, which catalyzes the dismutation of O_2_ to O^−2^ and H_2_O_2_. According to the current study, SOD activity was shown to be higher in wheat shoots and roots under As or Hg stress. Previously, under Hg stress, a similar SOD response was seen in *Chlamydomonas reinhardtii* [80]. Another study showed that SOD activity was elevated in various plant species under heavy metal stress, which validates our findings [81]. On the other hand, SOD activity was shown to be significantly reduced in wheat plants treated with As and Hg in combination with PDS, yet the decline in SOD level was extremely low when compared to control.

#### 3.6.4. PDS Induces CAT Activities under As and Hg Stress

Wheat seedlings treated with As and Hg showed the lowest CAT activity in roots/shoots, as compared to control and PDS treatment. It was recorded that the CAT activity in the root was 0.181 µM/g FW and the in shoot was 0.61 (treated with As). The CAT activity in the root was 0.185 µM/g FW and in shoot it was 0.149 µM/g FW (treated with Hg). However, the CAT activity in the root was 0.411 µM/g FW, and in shoot was 0.356 µM/g FW when treated with PDS. The CAT activity in the root was 0.372 µM/g FW and in the shoot was 0.310 µM/g FW in the control seedlings. In combination with Hg, it was found that PDS resulted in 0.237 µM/g FW, while in shoots, it showed 0.223 µM/g FW (treated with PDA + As). It was recorded that the CAT activity in the root was 0.226 µM/g FW and the shoot was 0.226 µM/g FW when treated with PDS + Hg (Figure 5G,H). Overall, the results show that As and Hg-treated plants significantly decreased CAT activity while PDS enhanced the CAT activities, and PDS combined with As and Hg reduced the CAT activities.

CAT is an antioxidant enzyme that has a high turnover rate and can prevent H_2_O_2_ formation [82]. Peroxidase and catalase are involved in the enzymatic degradation of H_2_O_2_ produced in cells when SOD activity is high [83]. There was a reduction in CAT activity when wheat seedlings were subjected to As and Hg stress. However, exposure to PDS enhanced the CAT activity in wheat plants. Previously, plants subjected to salt and heavy metal contamination increased their CAT activity, enabling them to scavenge H_2_O_2_ [84]. Lately, PDS has modulated the activity of POD, SOD, and APX under salt stress [43]. This might explain why wheat seedlings treated with PDS were more tolerant to salt exposure. Overall, the importance of PDS in improving cellular functions by boosting antioxidants and lowering H_2_O_2_ and lipid peroxidation during salt stress acclimation highlights the importance of maintaining redox homeostasis and preventing oxidative damage.

## 4. Conclusions and Future Recommendations

According to the outcomes of this study, heavy metal stress, such as As and Hg, brought about a significant decrease in the growth and biochemical activities of wheat plants. Particularly, the levels of H_2_O_2_, TBARS, and lipid peroxidation were significantly increased in wheat seedlings. On the other hand, PDS (2000 ppm) enhanced seed germination and the seedling growth of wheat. PDS, in combination with either As or Hg, alleviated the negative effects caused by As and Hg on wheat seedlings. Moreover, PDS significantly modulated the activity of antioxidants, such as POD, APX, SOD, and CAT, which resulted in a significant reduction in ROS (H_2_O_2_) and lipid peroxidation under As and Hg stress. This study concludes that PDS (2000 ppm) has the potential to alleviate the phytotoxic effects of As and Hg stress at the morphological and physiological level, which will lay a foundation for further understanding of the role of PDS in the regulation of different abiotic stress tolerance in plants. This study also provides a novel clue for farmers and nursery growers on how to promote the crop seedling’s growth under stressed conditions. It would be even more beneficial to analyze the effect of PDS and heavy metal stresses on the flowering and yield of wheat and other crops, in terms of the number and size of grain. It is also necessary to understand the molecular mechanism of how PDS alleviates the adverse effects of As and Hg stress in wheat.

## Figures and Tables

**Figure 1 plants-11-01379-f001:**
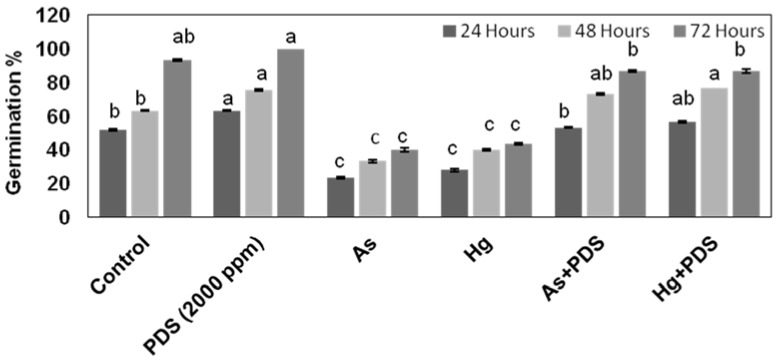
Effect of PDS (2000 ppm) on germination under normal (distilled water), PDS, As and Hg stress and combined application of As + PDS and Hg + PDS. Wheat seeds were subjected to distilled water (control), PDS, As, Hg, As + PDS, and Hg + PDS. Germination rate was observed on days 1–3. PDS (2000 ppm) enhanced seed germination rate and alleviated the adverse effects of As and Hg. *Y*-axis represents germination% while *X*-axis represents different treatments. All data were analyzed using one-way ANOVA, the different lowercase alphabets represent the homogenous subsets and it’s indicate the significantly according to Fisher LSD with multiple comparisons test at a significance threshold of *p ≤* 0.05 (*n* = 10). Each independent experiment was repeated three times.

**Figure 2 plants-11-01379-f002:**
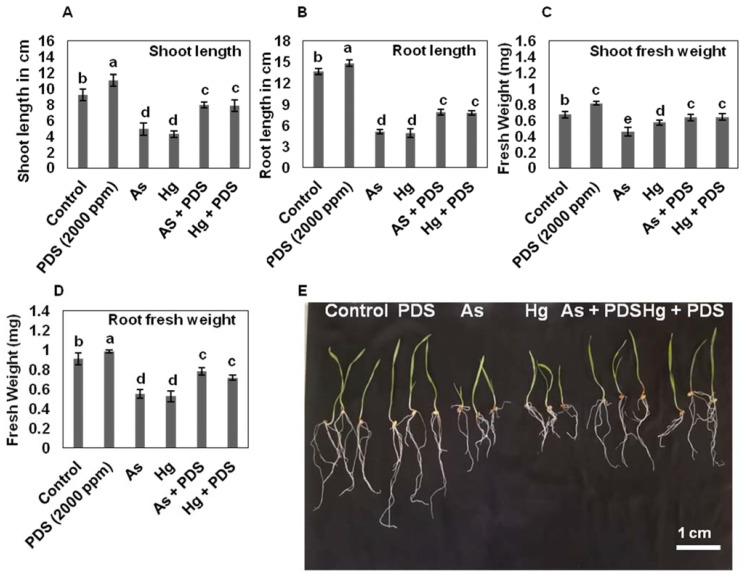
Effect of PDS (2000 ppm), As, Hg alone and in combined form on shoot/root length and fresh weight. The shoot (**A**) and root length (**B**) was measured under As, Hg, PDS, As + PDS and Hg + PDS. (**C**) Shoot (**C**) and root (**D**) fresh weight measured in mg. (**E**) Photograph was taken on day 7. Shoot/root length and fresh weight of ten randomly selected wheat seedlings from each treatment were recorded on day 7. The mean value was obtained from three different experiments performed at different intervals of time. All data were analyzed using one-way ANOVA, with multiple comparisons using LSD test at a significance threshold of *p ≤* 0.05 (*n* = 10). The different lowercase letters represent homogeneous subsets and indicate the statistical significance according to Fisher’s LSD at level *p ≤* 0.05.

**Figure 3 plants-11-01379-f003:**
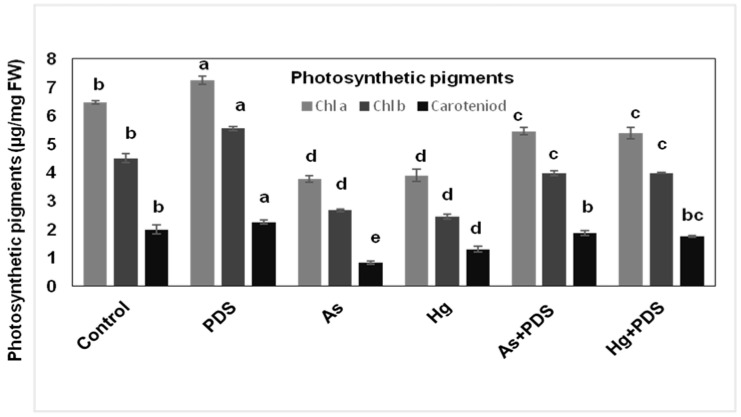
Evaluation of photosynthetic pigments under PDS (2000 ppm) and As and Hg-treated wheat seedlings. The photosynthetic activity was measured in wheat seedlings after 7 days of germination by selecting three random plants from each replicate and treatment. Chlorophyll a, b and carotenoids were increased under PDS alone and with As and Hg in combined form, but were decreased under As and Hg-treated plants. All data were analyzed using one-way ANOVA, with multiple comparisons using the LSD test at a significance level of *p ≤* 0.05 (*n* = 5). The different lowercase letters represent homogeneous subsets and indicate the statistical significance according to Fisher’s LSD at level *p ≤* 0.05.

**Figure 4 plants-11-01379-f004:**
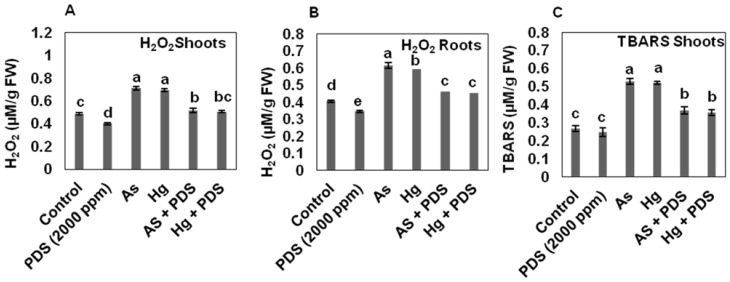
Effect of PDS (2000 ppm) on H_2_O_2_, lipid peroxidation and TBARS under normal, and As/Hg treatment in root ad shoot. The level of H_2_O_2_ was determined under control, PDS, As, Hg, As + PDS, and Hg + PDS in shoots and roots (**A**,**B**). Seven-day-old wheat seedlings were harvested and subjected to the analysis of H_2_O_2_ levels. TBARS and lipid peroxidation of 7-day-old seedlings were measured (**C**–**F**). Five seedlings were randomly collected from each replicate batch. All data were analyzed using one-way ANOVA, with multiple comparisons using the LSD test at a significance level of *p ≤* 0.05 (*n* = 5). The different lowercase letters represent homogeneous subsets that indicate the statistical significance according to Fisher’s LSD at level *p ≤* 0.05.

**Figure 5 plants-11-01379-f005:**
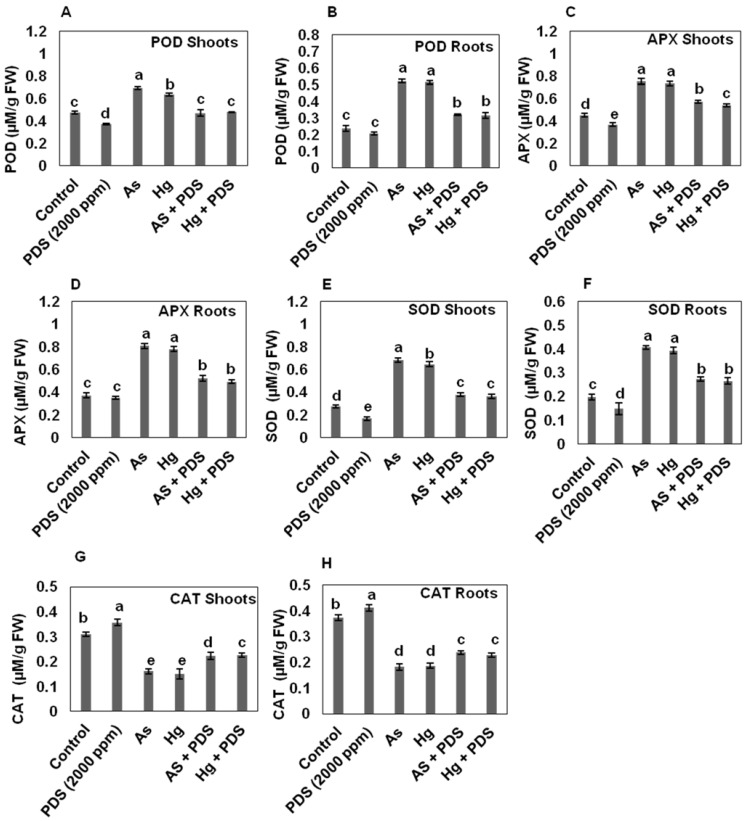
Evaluation of antioxidant enzymatic activity under PDS, As and Hg-treated wheat seedlings. Graphical bars (**A**,**B**) show POD in shoot and root, (**C**,**D**) show APX in shoot and root, (**E**,**F**) SOD and (**G**,**H**) CAT. Wheat seeds were treated with PDS (2000 ppm), As, Hg, PDS + AS and PDS + Hg. 7-day-old seedlings of wheat were harvested. Three randomly selected seedlings from each replicate were chosen for the analysis of POD, APX, SOD and CAT activities. The activities of POD, SOD and APX were lower under PDS treatment and higher under As and Hg treatment, while the CAT activity was higher under PDS and lower under As and Hg treatment. All data were analyzed using one-way ANOVA, with multiple comparisons using the LSD test at a significance level of *p ≤* 0.05 (*n* = 5). The different lowercase letters represent homogeneous subsets and indicate the statistical significance according to Fisher’s LSD.

## Data Availability

All related data are within the manuscript.

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
