# Peer review of "Plant-Derived Smoke Solution Alleviates Cellular Oxidative Stress Caused by Arsenic and Mercury by Modulating the Cellular Antioxidative Defense System in Wheat"

_plants, 2022, doi:10.3390/plants11101379_

Round 1

Reviewer 1 Report

Please double check grammar in the manuscript.  I saw a couple of grammatical issues.

Have you considered that leaves and roots contain endosymbiotic bacteria that may be involved in providing nutrients to plants.  Foliar fertilization with micronutrients stimulates growth of the bacteria in plant tissues.  The microbes themselves cause plants to increase expression of antioxidants and other oxidative stress responses.  Smoke may contain micronutrients and may be acting to feed these bacterial symbioses--causing increased microbial growth in plant tissues.  Root hairs and leaf trichomes are rich in the bacteria.  I don't know if you need to consider this in this paper--but the activation of symbiosis with microbes in plant cells and tissues may be an explanation.  Endophytic microbes cause the identical effects shown in smoke treatments.  Below is a paper from 2021 that describes this symbiosis in roots. 

Chang, X.; Kingsley, K.L.; White, J.F. Chemical Interactions at the Interface of Plant Root Hair Cells and Intracellular Bacteria. Microorganisms 2021, 9, 1041. https://doi.org/10.3390/microorganisms9051041 

Reviewer 2 Report

This is a competent study, with a clear outcome. Results look convincing and promising. The exact mechanism of regulation of stress response to heavy metals by PDS or its individual components (karrikins?) remains elusive, but this should not be a basis for the exclusion. However, it is highly desirable if As and Hg content measurements in different parts of plant, roots, shoots and leaves, should be undertaken. The problem is not only the reduced plant performance under the heavy metal stress, but the accumulation of toxic components in edible parts of plants.  Additional minor suggestions are as follows:

  1. Please, in the introduction after the line 58, consider the inhibitory effect of Hg on the plant aquaporins. Especially, keeping in mind the notion on the lines 68-69: "Hg can greatly affect plant essential activities such as... absorption of water,...". Suggested reference:  Protoplasma 2011 DOI: 10.1007/s00709-010-0222-9.
  2. Please, double check the units in which the H2O2 production and antioxidant enzymes activity is expressed, mM/gFW looks  suspicious for me.
  3.  Minor English issues, e.g. line 43 "are capable to be transported and accumulated"or simply "are transported and accumulated"; line 200, should be "distilled water"; line 204 should be full stop instead of comma after "Hg"; use more appropriate verb than "recorded" on line 463; "catalyzes" instead of "catalysis" on line 468; "imposed" instead of "posed" on line 511, etc.
  4. Correct multiple typos, like "Nest" on line 203, "pjeroxidase" on line 417, etc
  5. Change "chl" to "chlorophyll" on line 21. When you mention "chlorophyll a and b here and elsewhere (e.g. multiple entries in the section 3.3), use "and", not "& "
  6. Some figures diplay defects, e.g. Fig. 2, the PDS label is cut in the panel A, symbols d are superimposed with the error bars in the panel B, symbols are not centered with respect to bars in Figures 4 and 5, etc.
  7. Format of references in the reference list is invalid. 

Author Response

please see the below attachment
